# Feasibility and Reliability of Smartwatch to Obtain Precordial Lead Electrocardiogram Recordings

**DOI:** 10.3390/s22031217

**Published:** 2022-02-05

**Authors:** Nora Sprenger, Alireza Sepehri Shamloo, Jonathan Schäfer, Sarah Burkhardt, Konstantinos Mouratis, Gerhard Hindricks, Andreas Bollmann, Arash Arya

**Affiliations:** 1Department of Electrophysiology, Heart Center Leipzig, University of Leipzig, 04289 Leipzig, Germany; alireza.sepehri-shamloo@helios-gesundheit.de (A.S.S.); schaefer-jonathan@hotmail.de (J.S.); gerhard.hindricks@helios-gesundheit.de (G.H.); andreas.bollmann@helios-gesundheit.de (A.B.); arash.arya@helios-gesundheit.de (A.A.); 2Leipzig Heart Digital, Leipzig Heart Institute, 04289 Leipzig, Germany; konstantinos.mouratis@leipzig-heart.de; 3Institute of Therapy and Organizational Development, 10961 Berlin, Germany; sarah.burkhardt@gmx.net

**Keywords:** smartwatch, Apple Watch, electrocardiogram, feasibility, reliability, wearables, mobile health

## Abstract

The Apple Watch is capable of recording single-lead electrocardiograms (ECGs). To incorporate such devices in routine medical care, the reliability of such devices to obtain precordial leads needs to be validated. The purpose of this study was to assess the feasibility and reliability of a smartwatch (SW) to obtain precordial leads compared to standard ECGs. We included 100 participants (62 male, aged 62.8 ± 13.1 years) with sinus rhythm and recorded a standard 12-lead ECG and the precordial leads with the Apple Watch. The ECGs were quantitively compared. A total of 98 patients were able to record precordial leads without assistance. A strong correlation was observed between the amplitude of the standard and SW-ECGs’ waves, in terms of P waves, QRS-complexes, and T waves (all *p*-values < 0.01). A significant correlation was observed between the two methods regarding the duration of the ECG waves (all *p*-values < 0.01). Assessment of polarity showed a significant and a strong concordance between the ECGs’ waves in all six leads (91–100%, all *p*-values < 0.001). In conclusion, 98% of patients were able to record precordial leads using a SW without assistance. The SW is feasible and reliable for obtaining valid precordial-lead ECG recordings as a validated alternative to a standard ECG.

## 1. Introduction

The standard 12-lead electrocardiogram (ECG) is a frequently used diagnostic tool in cardiology [1]. It is a reliable method for diagnosing specific cardiac rhythm abnormalities, early assessment of cardiac ischemia, and can be used for progress monitoring in patients living with cardiac disorders [2,3]. It was also proven that the smartwatch obtains good results when monitoring the heart rate [4]. To successfully conduct a 12-lead ECG, it needs trained medical staff and a standard ECG device located in a hospital or an outpatient medical facility [5]. Obviously, it is time-consuming and complicated for patients to receive an ECG. 

Smartwatches (SWs) and smartphones are increasingly available worldwide and have a rapidly growing consumer base [6,7,8]. If a SW could obtain an accurate ECG recording, patients can quickly capture their ECG in case of any cardiac manifestation, before reaching a hospital or outpatient medical facility. This could save time, money, and the capacity of trained medical staff and increase the diagnostic yield. The Apple Watch from Series 4 can record a single-lead ECG with one negative and one positive electrode and received the de novo Food and Drug Administration clearance in 2018 [9]. Currently, the Apple Watch labels recorded ECGs as either sinus rhythm, high or low heart rate, atrial fibrillation, or inconclusive [9]. The results can be sent to a doctor via email and be analyzed. To incorporate the ECGs recorded by such devices in routine medical care, it must be ensured that the recordings are as reliable and valid as a standard 12-lead ECG.

During recent years, different research groups have tried to assess the feasibility and reliability of SWs to obtain ECG recordings. However, these attempts were mostly limited to studies that assessed qualitatively and quantitatively the similarity between the ECG recordings of the SWs and standard ECGs in the classical Einthoven ECG leads I-III or focused on the accuracy of the SW-ECG in pediatric patients with and without congenital heart disease [10,11,12]. A standard 12-lead ECG gives more information than lead I-III by recording the precordial leads V1–V6, and the Goldberger leads (aVR, aVL, and aVF) [13]. Although there are other studies that investigated the ability of a SW or other handheld devices to record precordial leads, the assessed sample sizes were limited to 3–52 patients or concentrated only on specific cardiac abnormalities such as ST-segment changes or atrial fibrillation [6,7,11,14,15,16,17,18,19]. Currently, some researchers have proposed methods to capture a standard 12-lead ECG with SWs and have shown a good concordance between Apple Watch ECGs and a standard 12-lead ECG; however, to the best of our knowledge, no large clinical study has assessed the reliability and validity of these methods. Our study examines the feasibility, reliability, and validity of a SW to obtain precordial leads compared to standard ECG in patients with sinus rhythm.

## 2. Materials and Methods

### 2.1. Study Design

This is a sub-study of the Leipzig Apple Heart Rhythm Study conducted in 2019–2020. Study participants were selected from patients referred to the Leipzig Heart Center. The Leipzig Apple Heart Rhythm Study is an observational, prospective study aiming to assess the feasibility and reliability of ECGs generated by SWs to diagnose cardiac arrhythmias. It is registered in ClinicalTrials.gov (NCT04092985). Patients participating in the study signed an informed consent form and were informed about their right to access their data and the right to withdraw from the study at any stage for any reason. The research project was approved by the Ethics Committee of the University of Leipzig. In the case of any research-related complications, patients were fully supported by researchers during the whole period of the study.

### 2.2. Study Participants

Participants with ages above 18 years and a sinus rhythm with no pathological findings in the 12-lead ECG were included. Patients with an implanted cardiac device, and mental or physical disability were excluded. Patient characteristics that were collected include age, gender, body mass index, medications, as well as presence or absence of ischemic heart disease, hypertension, diabetes, chronic obstructive pulmonary disease, and history of stroke and renal failure.

### 2.3. ECG Recording with Standard ECG

The 12-lead ECGs were recorded with two standard ECG devices (Edan SE1200 Express, CAmed Medical Systems GmbH, Leverkusen, Germany and CardioExpress SL12, Spacelabs Healthcare, OSI Systems, Inc., Snoqualmie, WA, USA). The parameters used were identical in both devices, paper running speed was 25 mm/s with an amplitude of 10 mm/mV. The low-frequency filtering was at 0.67 Hz, and the high-frequency filtering at 150 Hz. The patients were positioned in a supine position and asked to breathe spontaneously and avoid talking during ECG recording. The standard ECGs were evaluated directly after recording and patients meeting the inclusion criteria were included in the study.

### 2.4. ECG Recording with Apple Watch

Immediately after the standard ECG recording, the recording of SW-ECG with the Apple Watch was conducted without a change in body position. The Apple Watch has two electrodes; a positive one is on the back of the watch and a negative one on the Digital Crown. Patients were instructed how to record an ECG with the Apple Watch and positioned and taped the ECG by themselves with a research assistant present. The SW was placed in the same positions where the chest electrodes were placed for the standard ECG. To obtain V1, SW was placed in the 4th intercostal space to the right of the sternum, for V2 in the 4th intercostal space to the left of the sternum, for V3 it was placed diagonally between V2 and V4, V4 was placed between the 5th and the 6th rib in the midclavicular line, V5 on the same level as V4 in the anterior axillary line, and V6 on the same level as V5 in the midaxillary line ([Fig sensors-22-01217-g001]). The SW was held by patients at every place, with the back of the SW touching the skin and the right index finger touching the digital crown for 30 s to record a single lead ECG. In the standard ECG the precordial leads were recorded simultaneously, while in the SW-ECGs the recordings were sequential. In the standard ECG the unipolar leads are measured with respect to the Wilson central terminal while in the SW-ECG the right index finger is used. In the end, there were six single-lead ECGs per patient for V1–V6. The paper running speed in the Apple Watch was 25 mm/s with an amplitude of 10 mm/mV. These were transferred to the Health App of the iPhone 7 and then exported as PDF files. The Standard ECGs and the PDF files were anonymized and coded to be then analyzed.

### 2.5. Data Processing

The precordial leads from the standard ECG as well as the SW-ECG recordings were analyzed. Recordings were only used if the absence of baseline artifacts ensured good signal quality in >80% of the recording. Baseline artifacts were considered sections where no P-waves and/or no QRS-complexes could be identified. The heart rate was calculated from V1 of standard ECG and V1 from ECGs recorded with the Apple Watch, and the heart rate automatically calculated by the Apple Watch. In addition, a random heartbeat was selected and analyzed in the standard ECG from V1–V6 and in every lead of the Apple Watch ECG for the other variables. The variables used were the amplitude (in millivolts) of the P waves, QRS complexes, and T waves and the duration (in milliseconds) of the P waves, PR intervals, QRS complex duration, QT intervals, and T waves. The polarities (positive or negative) of the P waves, QRS complexes, and T waves ([Fig sensors-22-01217-g002]) were also assessed. Those measurements were completed with a standardized ECG ruler on the printed standard ECGs and SW-ECGs. Even though the standard ECGs differ in their appearance from the SW-ECGs when analyzed, the ECGs were randomized, so the analyzer could not know which standard ECG belonged to which SW-ECG.

### 2.6. Statistical Analysis

For reference, recordings of V1–V6 leads of the standard ECG were used. Continuous variables were expressed as mean ± standard deviation. Categorical variables were reported regarding numbers and proportions. To find the correlation between quantitative results of the ECG recording methods, the Pearson correlation test was used. For the qualitative data, Chi-square tests were used. The Pearson product-moment correlation coefficient (*r*) values were interpreted as follows: values *r* ≤ 0.20 as no agreement, 0.21–0.40 as weak, 0.41–0.60 as moderate, 0.61–0.80 as strong, and 0.81–1.00 as very strong correlation. A Bland–Altman analysis was used to calculate the agreement between the two measurement methods. The analytical software used in this study was SPSS version 17 (SPSS Inc., Chicago, IL, USA) and GraphPadPrism version 8.4.2. (San Diego, CA, USA). The two-tailed significance for *p* was set at <0.05.

## 3. Results

### 3.1. Baseline Characteristics

One hundred patients (62 male) with a mean age of 62.8 ± 13.1 years were enrolled. Table 1 summarizes the baseline patients characteristics.

### 3.2. Heart Rate

The mean heart rate calculated from the standard and SW-ECGs were 68.12 ± 12.54 and 69.25 ± 12.23 beats per minute, respectively (*r* = 0.96, *p* < 0.001). In addition, the heart rate calculated from the standard ECGs and the application report showed a strong correlation (*r* = 0.96, *p* < 0.001).

### 3.3. Amplitude, Duration, and Polarity of ECG Waves

A strong significant correlation was observed between the amplitude of the standard and SW-ECGs’ P waves, QRS-complexes, and T waves (Table 2). Moreover, a significant correlation was observed between the two methods regarding the duration of the ECG waves and intervals (Table 3). Assessment of polarity showed a significant and a strong concordance between the standard and SW-ECGs’ waves in all six leads (Table 4).

### 3.4. Ability to Record ECG

Out of 100 patients, only 2 needed assistance with the placement of the SW and the recording of the ECG. One patient had a tremor in the hand leading to shaking of the index finger on the digital crown and another patient had strong hair growth on the chest disturbing the contact of the SW to the skin. After a manual fixation of the shaking finger by the research assistant and the shaving of the chest hair in the other patient the ECGs could be recorded. In total, 98% of the participants had no problems using the SW for precordial lead ECG recording by themselves.

## 4. Discussion

### 4.1. Main Findings

There are three main findings in our study. First, 98% of the participants were able to record the SW-ECG by themselves without assistance. Second, the results showed that, with our described method, the SW could capture precordial recordings with similar reliability and validity compared to the standard ECG. Third, we found the best correlation for the P wave amplitude and visibility between the standard- and the SW-ECG in lead V2 (Table 2 and [Fig sensors-22-01217-g003]). Therefore, for interpreting the P wave especially in cardiac arrhythmias using SW-ECG we should focus preferably on lead V2 compared to other precordial leads.

### 4.2. Previous Studies

In the current study, we included only adult participants with sinus rhythm to validate SW-ECG precordial leads. However, in the other studies the focus was either on pediatric patients, or on the accuracy of SWs to capture cardiac arrhythmias such as atrial fibrillation in adults, the feasibility of obtaining the Einthoven leads I-III, and the ability of handheld devices to detect ST-segment changes [6,7,10,11,12,13,14,15,16,17,18,19].

Our study showed more potentials of a SW-ECG recording than detecting atrial fibrillation in just one or two leads. To the best of our knowledge, our study is one of the first large-scale studies to assess all Wilson Leads (V1–V6) with a SW in patients in sinus rhythm and quantitatively determined the similarity between a SW and a standard ECG.

Obtaining precordial leads using a SW has been reported in three studies so far [7,18,20]. This concept was initially described by Miguel Ángel Cobos. ECGs were obtained by using bipolar chest leads by placing one electrode on the right arm and the other on chest and were called CR1 to CR6 [7]. In the second study, a new terminology “Wilson-like” leads was used to define the positions of electrodes. In the study of Samol et al., patients had to hold their right wrist with their left hand; at the same time, the negative electrode must be touching the right finger to merge the right arm with left arm to create a new negative pole [20]. In this study, more than 90% of all Wilson-like SW-ECGs were correctly matched to the corresponding leads of a standard ECG. In the third study, 100 SW-ECGs by using the method described by Cobos (CR1 to CR6) were compared with the corresponding precordial V1–V6. However, the aim of this study was to identify the ST-segment changes in patients with acute coronary syndromes, and not the pericardial ECG waves’ characteristics. They finally showed a great agreement between the SW-ECG and standard ECG [18]. A study investigating the performance of handheld ECG devices to detect atrial fibrillation in a geriatric ward and a cardiology ward found that in 21.4% of the elderly patients in the geriatric ward, a recording with a handheld device could not be performed, while in the cardiology ward with younger patients, it was only 7% [21]. In another study by our research group, we found that patients had no problems with recording Einthoven leads I-III with a SW by themselves after a short training [10]. We assumed that it is more difficult for patients to record the precordial leads by themselves and position the SW at the right location than to record lead I-III, which only needs positioning of the SW on the wrist and the lower abdomen. However, after a short introduction, 98% of the participants of our study could position the SW by themselves and record the ECG independently. For the small percentage of patients who could not record a SW-ECG without assistance, there are already studies proposing solutions. They show how missing ECG leads can be synthesized from just one, three, or seven leads [22,23,24]. In our study, the exact positioning for the placement of the SW was always supervised by the researchers. However, for specific mobile applications, with the ability to instruct the patients with the exact positioning, measurements that can monitor the accuracy of the ECG recordings should be developed if we are going to extend the application of ECG recording by SWs. SW technology is enabling patients to record an ECG by themselves, no matter where they are, which could save resources for the patient as well as for the healthcare provider. However, these ECGs need to be interpreted either by an artificial intelligence algorithm or doctors who receive these ECGs in digital format. As mentioned before, the two major differences in the recording between the standard ECG and the SW-ECG were the simultaneous vs. consecutive recording and the Wilson central terminal vs. right index finger used for the measurement of the precordial leads. Despite these differences, we could confirm the reliability and validity of recorded SW-ECG precordial leads compared to standard ECG.

### 4.3. Future Perspective

Our results indicated that SW-ECG precordial leads using an Apple Watch have great potential to be used as an alternative to standard ECG devices in situations such as during a flight, in areas such as mountains far from medical facilities, or even at home, when patients feel chest discomfort or palpitations and want to immediately capture their ECG and share it with their physician for further diagnostic or therapeutic measures. However, to include SWs as a standard tool for cardiovascular diagnostic and progress monitoring of cardiovascular diseases, it needs to be ensured that the application is as easy and faultless as possible.

Recent studies evaluated the possibility of calculating the 12 ECG leads of a standard ECG by only recording 1, 3, or 7 leads, therefore paving the way for an even more simple and fast approach to obtain an ECG recording using a SW [22,23,24]. These studies, together with our findings, show the ability of a SW to record a validated 12-lead ECG, extrapolated based on a limited number of recorded leads.

Although in this study we showed the positioning of the electrodes for obtaining precordial leads, we believe that further investigations are still needed to test whether better positions can be found for obtaining higher quality ECGs. For example, for the three lead ECGs, initially the left lower electrode was placed on the ankle, however, later the proximal lower limb, which is nowadays known as the Lund system, was introduced as a position to reduce artefacts [25]. Placing the left lower electrode on the left torso was another approach which was used in different studies [26].

In the current study, the recording of SW-ECG with the Apple Watch was conducted without a change in body position immediately after the standard ECG recording. Therefore, the possibility of a simultaneous recording to be able to perform a 1:1 comparison between the two instruments was not evaluated. This can be investigated in further clinical studies.

Telemedicine is a constantly growing sector that offers an excellent opportunity to improve health care to become a more individual approach [27]. One aspect of this development is the availability of technology, allowing a more flexible approach to health care [28]. Our study proving the feasibility and reliability of a SW to obtain a precordial lead ECG shows that handheld ECG devices such as the SW could be a factor in this development. Our study was an attempt to highlight that this new technology can bear enormous potential in the future for a quickly available and uncomplicated method of monitoring cardiac activity in patients at risk and to diagnose conditions that might have been missed because of the effort needed to receive an ECG.

## 5. Limitations

The participants of this study were all in sinus rhythm and showed no signs of any arrhythmia on the ECG recordings. This was done to assess the concordance between the two measurement techniques. Nonetheless, further studies are needed to investigate the ability of a SW to recognize different cardiac arrhythmias or other pathologies that can be seen on an ECG. It should also be noted that during the recording of the SW-ECG, a research assistant was present to teach the participants, so we can make no statement about the quality of a SW-ECG obtained by patients without prior one-on-one instructions. Furthermore, patients were not asked to send the PDFs via email to a doctor by themselves, so we cannot say how well it works, especially for elderly patients. The frequency filtering parameters of the Apple Watch were not accessible for us. This could lead to differences in the morphology of the ECG waves on the SW-ECG compared to the Standard ECG. The device used in this study was an Apple Watch Series 4, so the results cannot be transferred to other handheld devices.

## 6. Conclusions

Our study showed the feasibility, reliability, and validity of a SW to obtain precordial lead ECG recordings. A significant strong correlation could be observed between the SW and the standard ECG’s measured variables. Although all obtained precordial leads were suitable for analysis of QRS complexes and T waves, V2 was the best lead for analysis of the P waves in a SW-ECG. Finally, most of the patients (98%) were able to record precordial leads using a SW without assistance after a short one-on-one instruction and training. As further steps, we suggest performing studies in patients with different types of cardiac arrhythmia and ECG morphology changes. Since this study was limited to assessing the concordance between the two measurement techniques in patients with sinus rhythm, further studies are needed to investigate the ability of a SW regarding ECG changes in the setting of different cardiac disorders that can be reflected on an ECG. Moreover, our results are limited to one specific kind of smart watch, therefore, for using other smart watches in clinical practice further studies are warranted.

## Figures and Tables

**Figure 1 sensors-22-01217-g001:**
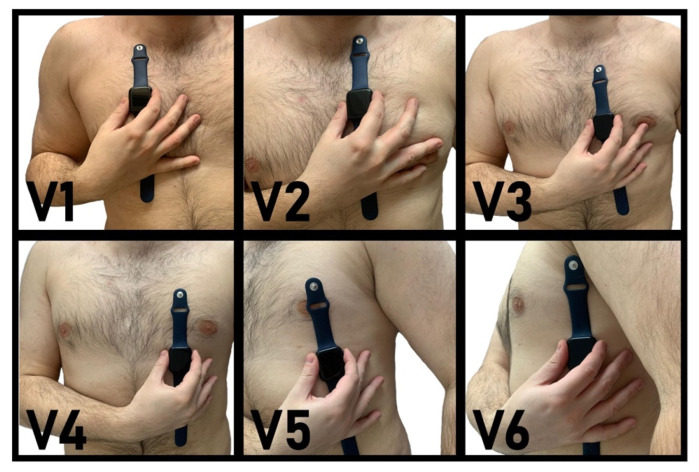
The positioning of the Apple Watch Series 4 for obtaining precordial leads. In the Apple Watch, the negative electrode is placed in the digital crown and the positive electrode is on the back crystal of the watch. The patient is in supine position.

**Figure 2 sensors-22-01217-g002:**
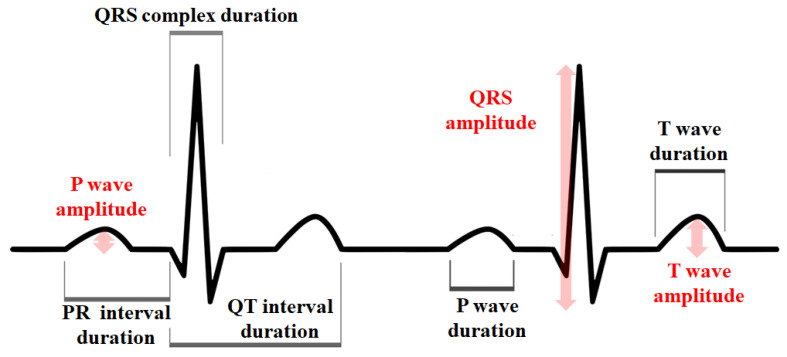
Electrocardiogram characteristics used for the assessment.

**Figure 3 sensors-22-01217-g003:**
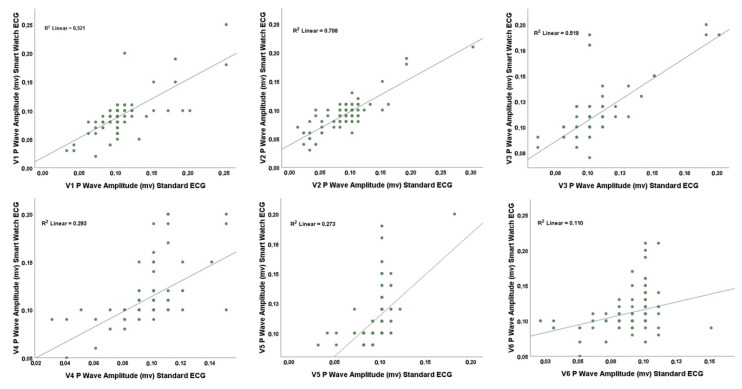
Correlation between P wave amplitude in precordial leads between standard- and SW-ECG.

**Table 1 sensors-22-01217-t001:** Patient baseline characteristics.

Variables	Number of Patients
Age (year), mean ± SD	62.79 ± 13.05
Gender (male/female), number	62/38
Body mass index (kg/m^2^), mean ± SD	27.35 ± 4.72
Ischemic heart disease	27
Hypertension	72
Diabetes	31
Prior stroke	4
Renal failure	14
Chronic obstructive pulmonary disease	13
Beta-blocker	56
Digoxin	0
Amiodarone	3
Angiotensin receptor blocker (ARB)	34
Angiotensin-converting enzyme-inhibitor (ACEI)	33
Antiplatelet drug	39
Anticoagulant	25

**Table 2 sensors-22-01217-t002:** Electrocardiogram (ECG) characteristics (amplitude).

Variables, Units		Lead	Standard-ECG	SW-ECG *	Correlation Coefficient	*p*-Value	Bias (95% LoA) **
**Amplitude (millivolts)** **Mean ± SD**	**P wave**	V1	0.10 ± 0.03	0.09 ± 0.03	0.72	<0.001	0.01
		V2	0.09 ± 0.04	0.09 ± 0.03	0.84	<0.001	0.00
		V3	0.10 ± 0.02	0.11 ± 0.02	0.72	<0.001	−0.01
		V4	0.10 ± 0.02	0.11 ± 0.03	0.54	<0.001	−0.01
		V5	0.09 ± 0.02	0.11 ± 0.02	0.52	<0.001	−0.02
		V6	0.09 ± 0.02	0.11 ± 0.03	0.33	<0.01	−0.02
	**QRS complex**	V1	0.92 ± 0.41	0.58 ± 0.31	0.70	<0.001	0.34
		V2	1.25 ± 0.52	1.07 ± 0.55	0.82	<0.001	0.18
		V3	1.37 ± 0.60	1.40 ± 0.59	0.87	<0.001	−0.03
		V4	1.43 ± 0.63	1.67 ± 0.57	0.88	<0.001	−0.24
		V5	1.42 ± 0.57	1.75 ± 0.55	0.88	<0.001	−0.25
		V6	1.16 ± 0.46	1.55 ± 0.47	0.82	<0.001	−0.39
	**T wave**	V1	0.13 ± 0.08	0.12 ± 0.07	0.81	<0.001	0.01
		V2	0.24 ± 0.17	0.28 ± 0.17	0.84	<0.001	−0.04
		V3	0.24 ± 0.16	0.28 ± 0.17	0.91	<0.001	−0.04
		V4	0.24 ± 0.16	0.30 ± 0.18	0.93	<0.001	−0.06
		V5	0.20 ± 0.12	0.29 ± 0.15	0.89	<0.001	−0.09
		V6	0.19 ± 0.11	0.25 ± 0.12	0.84	<0.001	−0.06

* Smartwatch (SW)-ECG; ** Bland–Altmann analysis and 95% limits of agreement.

**Table 3 sensors-22-01217-t003:** ECG characteristics (duration, mean *±* SD).

Variables, Units		Lead	Standard-ECG	SW-ECG *	Correlation Coefficient	*p*-Value	Bias (95% LoA **)
**Duration (milliseconds)**	**P wave**	V1	91.35 ± 12.85	93.40 ± 11.54	0.84	<0.001	−2.05
		V2	91.60 ± 11.26	94.34 ± 11.35	0.86	<0.001	−2.74
		V3	94.50 ± 11.04	98.99 ± 10.93	0.80	<0.001	−4.49
		V4	98.30 ± 11.46	101.43 ± 11.12	0.88	<0.001	−3.13
		V5	98.30 ± 11.90	100.51 ± 12.15	0.87	<0.001	−2.21
		V6	100.60 ± 12.78	101.68 ± 12.03	0.91	<0.001	−1.08
	**PR interval**	V1	162.60 ± 32.24	163.12 ± 32.94	0.97	<0.001	−0.52
		V2	159.60 ± 34.43	162.53 ± 29.70	0.85	<0.001	−2.93
		V3	166.60 ± 31.95	164.75 ± 30.18	0.92	<0.001	1.85
		V4	167.00 ± 32.80	164.69 ± 30.47	0.92	<0.001	2.31
		V5	166.80 ± 31.65	165.56 ± 29.91	0.90	<0.001	1.24
		V6	166.10 ± 32.60	164.29 ± 27.92	0.89	<0.001	1.81
	**QRS complex**	V1	97.60 ± 15.12	98.65 ± 14.98	0.95	<0.001	−1.05
		V2	98.60 ± 14.00	100.91 ± 16.11	0.85	<0.001	−2.31
		V3	99.65 ± 12.48	102.47 ± 14.65	0.82	<0.001	−2.82
		V4	100.00 ± 14.63	102.04 ± 15.13	0.95	<0.001	−2.04
		V5	96.10 ± 15.76	99.90 ± 16.63	0.82	<0.001	−3.8
		V6	94.20 ± 14.01	98.67 ± 13.29	0.83	<0.001	−4.47
	**QT interval**	V1	385.40 ± 33.10	386.56 ± 34.18	0.88	<0.001	−1.16
		V2	386.16 ± 37.22	382.76 ± 36.29	0.85	<0.001	3.4
		V3	404.70 ± 39.55	402.42 ± 38.65	0.98	<0.001	2.28
		V4	405.10 ± 38.60	404.59 ± 37.53	0.97	<0.001	0.51
		V5	399.40 ± 39.92	388.54 ± 36.68	0.88	<0.001	10.86
		V6	399.80 ± 39.82	390.61 ± 39.14	0.86	<0.001	9.19
	**T wave**	V1	160.20 ± 22.25	163.75 ± 20.74	0.81	<0.001	−3.55
		V2	172.02 ± 29.38	174.59 ± 25.61	0.95	<0.001	−2.57
		V3	177.50 ± 23.67	178.28 ± 22.81	0.95	<0.001	−0.78
		V4	179.60 ± 27.04	181.02 ± 25.06	0.95	<0.001	−1.42
		V5	153.90 ± 28.46	164.95 ± 24.26	0.81	<0.001	−11.05
		V6	169.00 ± 27.65	173.47 ± 24.16	0.94	<0.001	−4.47

* SW-ECG; ** Bland–Altmann analysis and 95% limits of agreement.

**Table 4 sensors-22-01217-t004:** ECG characteristics (polarity).

Variables		Lead	Standard-ECG	SW-ECG *	Concordance (%)	*p*-Value
**Polarity** **(positive/negative)**	**P wave**	V1	48/49	49/48	99%	<0.001
		V2	75/24	77/22	98%	<0.001
		V3	100/0	99/0	100%	<0.001
		V4	100/0	98/0	100%	<0.001
		V5	100/0	99/0	100%	<0.001
		V6	100/0	98/0	100%	<0.001
	**QRS complex**	V1	6/91	7/90	99%	<0.001
		V2	14/85	21/78	93%	<0.001
		V3	20/79	25/74	93%	<0.001
		V4	63/35	66/32	97%	<0.001
		V5	85/14	87/12	98%	<0.001
		V6	92/6	93/5	99%	<0.001
	**T wave**	V1	44/53	53/44	91%	<0.001
		V2	85/14	88/11	97%	<0.001
		V3	86/13	86/13	98%	<0.001
		V4	87/11	87/11	100%	<0.001
		V5	86/13	86/13	100%	<0.001
		V6	88/10	88/10	100%	<0.001

* SW-ECG.

## Data Availability

The anonymized data presented in this study are available on request from the corresponding author after authorization from the institutional data protection committee. The data are not publicly available due to the patient privacy policy.

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
