# Peer review of "Feasibility and Reliability of Smartwatch to Obtain Precordial Lead Electrocardiogram Recordings"

_sensors, 2022, doi:10.3390/s22031217_

Round 1

Reviewer 1 Report

The authors make an interesting study on the feasibility of the apple watch. This is an important study for the heart rate-related wearable community. Findings are well discussed. 

  1. There are many acronyms in this study, a separate table for acronyms would increase readability.
  2. Conclusion section is very short. There are many important findings, these findings need to be better highlighted.
  3. Related work can be extended, in this current form it misses many important studies.
  4. Table 1 needs to be reorganized and explained since it is not intuitive for some metrics. For example, what does Hypertension = 72 means?

References:

  1. Phan, D., Siong, L. Y., Pathirana, P. N., & Seneviratne, A. (2015, October). Smartwatch: Performance evaluation for long-term heart rate monitoring. In 2015 International symposium on bioelectronics and bioinformatics (ISBB) (pp. 144-147). IEEE.

Reviewer 2 Report

The paper presents a thorough study on the feasability of using a smart watch (in particular Apple Smart Watch) to record precordial ECG.

The study is well structured and the experimental data are carefully analyzed.

The drawn conclusions are solid.

I only have a few suggestions on some aspects that might be worth investigating. Of course these aspects might well be the object of future studies by the authors, but I think they  deserve at least a brief mention in the present manuscript.

  1. Based on the obtained results, which show that the P wave amplitude cannot be reliably detected on all precordial leads, but mainly on V2, it would be interesting to highlight which is the diagnostic potential for these SW recordings. Could they be a substitute of a standard 12-lead ECG for all kinds of heart conditions or due to the limitations of the SW recordings some specific pathological conditions might be difficult to detect?
  2. The present study records the precordial leads with the SW after the recording with the standard device. This only allows a "statistical" comparison between the two devices. Did the authors assess the possibility of a simultaneous recording in order to be able to perform a 1:1 comparison bewteen the two instruments?
  3. The study was limited to one specific kind of smart watch. Do the authors have any experience to predict how much difference one could expect bewteen one kind of smart watch and another? Would they all be equally suitable for the task?

Reviewer 3 Report

This manuscript presents a new study on the usage of Apple watches to register electrocardiograms (ECG). The idea is to determine a procedure to capture patient status by using this device, compare it with standard appliances and identify the real possibilities of the SmartWatch (SW). As the authors said, the Apple Watch can record single-lead ECG, and they can validate the reliability of such devices to obtain precordial leads before incorporating SW in routine medical care. The research included 100 participants with sinus rhythm and recorded a standard 12-lead ECG and 15 precordial leads with the Apple Watch. The ECGs were quantitively compared. The authors identified a strong correlation between the amplitude of the standard and SW-ECGs' waves. Additionally, a significant correlation was observed between the two methods regarding the duration of the ECG waves. In conclusion, the authors affirm that the SW is feasible and reliable to obtain valid precordial-lead ECG recordings as a validated alternative to a standard ECG.

The authors did a good job. The experiment is well described and will certainly advance the state-of-the-art in this specific application of sensors in healthcare. Despite that, I have some suggestions that may improve the final version of this manuscript.

  • I missed a table synthesizing the main desired characteristics of the SW ECG observed in the related publications.
  • An in-depth evaluation of the main proposals would enrich the present manuscript.
  • Another table comparing the obtained results with others already published would help users understand how far the authors' proposal is.

The suggestions are not mandatory. I enjoyed reading this manuscript and believe it will contribute to many other groups researching the same topic.
